# New Self-Report Measures of Commuting Behaviors to University and Their Association with Sociodemographic Characteristics

**DOI:** 10.3390/ijerph182312557

**Published:** 2021-11-29

**Authors:** Ximena Palma-Leal, Fernando Rodríguez-Rodríguez, Pablo Campos-Garzón, Antonio Castillo-Paredes, Palma Chillón

**Affiliations:** 1PROFITH Research Group, Department of Physical Education and Sports, Faculty of Sport Sciences, Sport and Health University Research Institute (iMUDS), University of Granada, 18011 Granada, Spain; ximena.palmaleal@gmail.com (X.P.-L.); pcampos@ugr.es (P.C.-G.); 2IRyS Group, School of Physical Education, Pontificia Universidad Católica de Valparaíso, Viña del Mar 2340000, Chile; fernando.rodriguez@pucv.cl; 3Grupo AFySE, Investigación en Actividad Física y Salud Escolar, Escuela de Pedagogía en Educación Física, Facultad de Educación, Universidad de Las Américas, Santiago 8370035, Chile; acastillop85@gmail.com

**Keywords:** commuting behaviors, physical activity, university students, active transport

## Abstract

Active commuting provides an opportunity for increased physical activity levels by a simple, inexpensive, and easy way to be incorporated in daily routines and could be considered a steppingstone for achieving a sustainable society since it provides physical, psychological, environmental, and economic benefits. Objective: (a) to describe the commuting patterns to and from university in students regarding gender, (b) to provide new self-report variables to measure the active commuting behavior, and (c) to examine the sociodemographic characteristics associated with commuting behaviors. Material and Methods: A total of 1257 university students (52.4% females) participated (22.4 ± 5.6 years old) from three Chilean universities located in different cities. Results: 56.1% of women and 42.0% men use public bus to and from university. The commuting energy expenditure was higher in active commuting followed by public and private modes of commuting (*p* < 0.001). The most active commuters were those older (men: OR = 3.637; 95% CI = 1.63, 8.10; women: OR = 8.841; 95% CI = 3.94, 13.78), those who lived in university residence (men: OR = 12.432; 95% CI = 4.39, 35.19; women: OR = 3.952; 95% CI = 1.31, 11.85), belonged to low socioeconomic level (men: OR = 3.820; 95% CI = 1.43, 10.18; women: OR = 4.936; 95% CI = 1.63, 14.90), and to public universities (men: OR = 26.757; 95% CI = 10.63, 67.34; women: OR = 8.029; 95% CI = 3.00, 21.48). Conclusion: The sociodemographic characteristics may influence in the mode of commuting to university. New variables of commuting behaviors may be efficient to quantify the physical activity.

## 1. Introduction

Physical inactivity is the fourth leading cause of death worldwide and was described as a pandemic that needs urgent action [1]. Moreover, it has been recognized as a major risk factor for non-communicable diseases [2]. It has evidenced that physically inactive adult people are 20–30% times more likely at risk of death compared to active people [3].

Physical activity (PA) promotion is a priority across different ages, social groups, and countries worldwide [4]. University students could be a key population for PA promotion [5] since the adopted behaviors in this period play a role in the consolidation of patterns throughout their life spans [6,7]. However, this period might be accompanied by the abandonment of healthy routines and habits established [8]. Active commuting, such as walking or cycling, provides an opportunity for increased PA levels [9,10,11] through a simple, inexpensive, and easy way to be incorporated in daily routines [12]. Furthermore, this behaviors provides psychological [13], environmental [14], and economic [15] benefits. In fact, active commuting to and from university has been showed to be a useful strategy to increase daily PA levels [16]. In addition, active commuting could be considered a steppingstone for achieving a sustainable society [17]. However, a limited number of studies have been conducted on university student populations.

The research on the commuting behaviors do not present a consensus on how to evaluate this properly. Most of the studies focused on active commuting use self-reported measures, and the categorical variable of active vs. passive mode of commuting is mostly used and included in the statistical analysis [18]. For instance, self-reported measurers could be associated with recall bias and social desirability [19,20]. On the other hand, it has been declared that the research of commuting behaviors may capture valued information but omit important differences if objective measures are not evaluated [21]. Therefore, an appropriate approach is to consider that objective and self-reported measures do not have to substitute each other, and both should only be seen as complementary instruments [22]. Several studies recommend the inclusion and accurately assess of dose and measure variables to improve the comparability of data and provide a better perspective [23,24].

The active commuting to university behaviors may be influenced, following the social-ecological model since childhood [25], by individual, social, community, environment, and policy factors. However, little is known about specific factors influencing the active commuting to university, and more evidence is needed [26]. Examining the individual sociodemographic characteristics related to active commuting to university provides an empirical basis for interventions that could be implemented by universities to increase active commuting [27]. University students in the United States indicated that the mode of commuting choice is influenced by their type of residence, but other types of information may affected their mode of commuting choice [28]. Similarly, in Brazil, active commuting was associated with low income in work population [29], but university students were not studied. In Chile, the university students revealed that personal factors were more influencing than environmental aspects to using active commuting [30].

To the best of our knowledge, there is a demanding need to explore new and quantitative self-report measuring active commuting. Moreover, little evidence is available related to the commuting behaviors to university and its association with sociodemographic characteristics. Therefore, the aims of the current study were (a) to describe the commuting patterns to and from university in students regarding gender, (b) to provide new self-report variables to measure the active commuting behavior, and (c) to examine the sociodemographic characteristics associated with commuting behaviors.

## 2. Materials and Methods

### 2.1. Study Design and Participants

This cross-sectional and non-randomized study was conducted between April and November of 2017. A total of 1257 university students (52.4% women) with an average age of 22.4 ± 5.6 years participated in this study. The participants were recruited from three different public and private universities located in different cities (two in Valparaíso and one in Santiago). The students belonged to diverse faculties (art, engineering, health, social sciences, and education), and the range of stay of the university students was from one to ten semesters.

### 2.2. Procedures and Ethical Requirements

Firstly, a letter was sent to the corresponding authorities of the different universities explaining the objectives of the study. Once the authorization was obtained by the authorities, all university students that voluntarily agreed to participate received information about the project and filled out an informed consent to participate in this study. The informed consent explained the characteristics of the questionnaire, the purpose of the study, and the confidentiality of the results. The students participated by completing a 15- to 30-min self-reported paper-based questionnaire implemented by volunteer teachers that were previously trained. All procedures followed the Helsinki protocols [31] and were approved by the Ethics Committee of Pontificia Universidad Católica de Valparaíso (Code: CCF02052017).

### 2.3. Instruments

The self-reported questionnaire used was created at the School of Physical Education of the Pontificia Universidad Católica de Valparaíso by researchers on the topic of active commuting. The questionnaire called “Questionnaire of mode of commuting and PA to the university” has a total of 28 questions and was created after a deep literature review and expert’s consultation. This includes questions about sociodemographic variables and commuting behaviors. Every question was adapted to Chilean university students’ context and has been reliable for university students in Chilean university students [32].

#### 2.3.1. Sociodemographic Characteristics

Participants reported their name, age, gender, year of admission to the university, postal address (as university student), type of residence (e.g., family residence or university residence), locality area (e.g., urban, or rural), and socioeconomic characteristics. The age was classified in two categories: younger (18–25 years old) and older (>26 years old) [33]. The type of residence was assessed using the question: With whom do you live? Answer options were divided into two categories: family residence (e.g., parents’ home or own house) and university residence (e.g., shared flat with other students or hall of residence), as it has been reported in previous studies [34]. The locality area was assessed with the question: Where is the area you reside as a student? Answer options were two categories: urban and rural. The family affluence scale (FAS) for socioeconomic levels was used [35]. The variables of family housing conditions are defined with the following questions: Does your family own a car? (No (0); Yes, one (1); Yes, two or more (2)); How many computers does your family own? (None (0); One (1); Two (2); More than two (3)); Do you have your own bedroom for yourself? (No (0); Yes (1)); and Do you have internet access? (No (0); Yes (1)). A score was assigned to each answer and then summed in order to obtain the total points (from 0 to 7 points) [36]. Therefore, participants were classified into three categories regarding the socioeconomic status levels: low (0 to 3 points), medium (4 to 5 points), and high (6 to 7 points). Finally, the type of university matches according to the type of the geographic area and were classified in two categories: public (Valparaíso) and private (Santiago). Although both cities are in the central zone of Chile, Santiago is located in a valley surrounded by the Andes and Chilean Costal mountains, and Valparaíso is a port city on the coast of Chile.

#### 2.3.2. Commuting Behaviors

##### Mode of Commuting

The question about mode of commuting was suggested in a systematic review of 158 studies in the scientific literature and has been proposed as the most appropriate measurement for asking about the mode of commuting [18]. The question about mode of commuting used in the current study was validated [37] and reliable [38] in young Spanish people and has been reliable for university students in Chilean university students [32]. The mode of commuting to and from university was assessed using separate questions: How do you usually travel to and from university? The answer options were: walking, cycling, car, motorcycle, public bus, metro/train, and others. Participants were classified in three categories as: active (walking and cycling), private (car and motorcycle), and public (public bus and metro/train) commuting, similar to previous studies [39]. Students who answered *combined* (e.g., active + private) were classified in the mode of commuting involving the highest PA levels. Active commuting involves the highest PA levels, followed by public commuting, which involves intermediate level of PA by walking to and from stations and stops, and private commuting, assumed to involve the lowest PA levels [40]. 

##### Commuting Time, Commuting Distance, Commuting Speed, and Commuting Energy Expenditure

Time and distance were assessed and based on this information, and two variables were calculated to provide further insights into the commuting behaviors: speed and energy expenditure. The commuting time to and from university was assessed using the question: How long does it take to go from home to university? Participants indicated the minutes per day dedicated to each mode of commuting to and from university from Monday to Friday [32]. These minutes were separated according to the category of mode of commuting. The commuting distance to and from university was assessed using the question: What is your postal address where you live as student? Once the university students self-reported their postal address, the research team geocoded both home and university in Google Maps, selecting the shortest network distance on foot in kilometers (km), as it has been reported in previous studies [41]. The commuting speed was calculated according to the equation: commuting distance in km divided by commuting time in hours (the minutes of commuting time were previously converted into hours), and the result was expressed in km/hour. The commuting energy expenditure expressed in Metabolic Equivalents (METs) was calculated to estimate the energy cost of each mode of commuting (per minute), based in the code of Compendium of Physical Activities for adults [42]. The specific METs score assigned (per minute) by each mode of commuting according to the code in the Compendium was multiplied by the commuting time to obtain the total energy expenditure estimated (see calculation examples, Appendix A). These two commuting energy expenditures (per minute and total) were calculated to differentiate the modes of commuting since it is possible that total commuting energy expenditure from two different participants may be similar, but these expenditures might come from different behaviors (e.g., walking vs. sitting in public commuting), and consequently, commuting energy expenditure per minute might be different. In active commuting, the energy expenditure per minute was established with their respective commuting speed according to the ranges of the Compendium. The METs used in this study were: (i) 2.0 METs (less than 3.0 km/h—code 17151); (ii) 2.8 METs (3.0 to 3.9 km/h—code 171152); (iii) 3.0 METs (4.0 to 4.49 km/h—code 17170); (iv) 3.65 METs (4.5 to 5.49 km/h—code [17170 + 17200]/2); (v) 4.3 METs (5.5 to 6.49 km/h—code 17200); (vi) 5 METs (6.5 to 6.9 km/h—code 17220); (vii) 7 METs (7 to 8.49 km/h—code 17230); and (viii) 8.3 METs (≥8.5 km/h—code 17231). A previous study calculated the estimation of energy expenditure in active commuting [43] by multiplying the METs score of the Compendium by the minutes per week spent walking and cycling but did not establish commuting speed. Therefore, in the current study, we sought higher accuracy for the data obtained. For public and private commuting, the commuting energy expenditure per minute was not assigned according to the commuting speed calculated since these speeds correspond to motorized transport. For public and private commuting, the METs used were 1.3 METs (code 16016, riding in bus or train and code 16015, riding in car, respectively). Additionally, for public commuting, previous research has determined that the median walking time included in this type of commuting is 15 min per day [40]. Therefore, 7.5 min per trip (to and from university) with an energy expenditure of 2.5 METs (code 17161), as considered in the Compendium for walking to the stations and stops [42], was used.

### 2.4. Statistical Analysis

Mode of commuting and sociodemographic characteristics were analysed using descriptive statistics and were reported as the mean ± standard deviation (SD) for continuous variables and as frequencies and percentages (%) for categorical variables. The significant differences in these descriptive variables for men and women were analysed using chi-square test for categorical variables and standard analysis of variance (ANOVA) for continuous variables, where the level of significance was set to *p* < 0.05. Associations between mode of commuting with sociodemographic characteristics were studied using multinomial logistic regression analysis. Mode of commuting were included in the model as the dependent variable, and sociodemographic characteristics were included as independent variables individually and adjusted by distance, age, and socioeconomic levels (except in the analysis when that variable was the predictor). Associations between commuting behaviors (speed and energy expenditure) and sociodemographic characteristics were studied using linear regression. Commuting behaviors were included in the model as the dependent variable, and sociodemographic characteristics were included as independent variables individually and adjusted by distance, age, and socioeconomic levels (except in the analysis when that variable was the predictor). The statistical analyses were conducted using the IBM SPSS Statistics (v. 25.0 for WINDOWS, Chicago, IL, USA), and all analyses were performed jointly for men and women and adjusted by gender.

## 3. Results

The commuting patterns to and from university by gender are shown in Figure 1. The main mode of commuting to and from university was public bus, which was higher in women than men (*p* < 0.001). The second mode of commuting most used was walking, with men showing higher percentages than women (*p* < 0.001). Two statistical differences in the mode of commuting from university by car and cycling were found, with men showing lower percentages in car (*p* < 0.05) and higher percentages in cycling than women (*p* < 0.05). Regarding going to university by motorcycle, there was significant differences between women and men (*p* < 0.05). 

Sociodemographic characteristics and commuting behaviors by gender are presented in Table 1. The mean age was 22.4 ± 5.6 years; men were significantly older than women (*p* < 0.001), and more men than women lived in a family residence (*p* < 0.001). Most of the sample lived in an urban area (96.8%) and had a medium socioeconomic status (52.6%), without statistical differences among genders. A percentage of 71.4% men and 44.9% women came from public universities with geographical area in Valparaíso (*p* < 0.001). The commuting time and distance to university were higher in public than private mode of commuting (*p* < 0.001). The commuting speed was higher in private mode of commuting followed by public and active mode of commuting (*p* < 0.001). The commuting energy expenditure per min was higher in active commuters followed by public and private commuters, and total commuting energy expenditure was higher in public commuters, followed by active and private commuters in women and men (all, *p* < 0.001).

The mode of commuting associated with sociodemographic characteristics by gender are presented in Table 2. The most active commuters were those older, those who lived in university residence, and belonged to public university from Valparaíso (*p* < 0.05). In addition, the women that belonged to low and medium socioeconomic level also showed to be the most active commuters (*p* < 0.05). On the other hand, men who used public mode of commuting were older, belonged to low socioeconomic level, and to public universities from Valparaíso and women who used public mode of commuting were those who belonged to low and medium socioeconomic level, and to public universities from Valparaíso compared to those who used private mode of commuting (*p* < 0.05). According to the commuting behaviors, in men and women, active commuting decreased with higher commuting time and commuting distance to university (both, *p* < 0.05). Finally, active and public commuters reported higher energy expenditure per min and total energy expenditure compared to those that use private mode of commuting (both, *p* < 0.05).

The commuting behaviors to university associated with sociodemographic characteristics by gender are presented in Table 3. The commuting speed presented negative associations with students who lived in university residence, both in men and women (both, *p* < 0.05). In addition, the commuting speed to university showed one negative association with male students who belong to public university from Valparaíso and one positive association with women who belonged to urban locality area (both, *p* < 0.05). Regarding commuting energy expenditure per min, men and women presented positive associations with students who lived un university residence and belonged to public university (both, *p* < 0.05).

## 4. Discussion

The main findings were that the public mode of commuting was the most used by Chilean university students; secondly, new variables of commuting behaviors may be efficient to quantify the PA, and thirdly, Chilean students were more likely to use active and public mode of commuting to university when they belonged to public universities from Valparaíso, lived in university residence, were older students, and had a low socioeconomic level.

According to commuting patterns to and from university, the public bus was the main mode of commuting to and from university in men and women. Similar results were reported in Canada, where the majority of university students (55.0%) used public bus as a preference [44]. Walking was the second mode of commuting to and from university in men and women, such as in a study in Spain [43], where 22.3% of men and women also reported walking as the second mode of commuting. Notwithstanding the similar results, men showed lower percentages in the use of public bus and higher percentages in walking than women. In the United States, it was found that gender tends to affect the choice of the mode of commuting of university students [45], and in Chile, the mode of commuting of women is determined by the security that this implies for them [46]. Further, in the current study, men present a higher percentage in cycling and motorcycle than women, being the less common modes of commuting to and from university in both genders. Regarding to cycling, similar results were obtained in a study in Costa Rica, where approximately 1% of university students use the bicycle commuting to and from university [47]. Regarding to motorcycle, it has been reported that the majority of users of motorcycles in Latin America countries, such as Brazil, Colombia, Argentina, and Venezuela, were mainly young men [48], but lack of studies related to use of motorcycles in university students makes difficult the comparison with the results of the present study. In the same way, the most likely cause of this result may be due to the fact that across Latin America, the infrastructure or road safety conditions for active commuting are generally poor and/or non-existent and discourage potential users [49]. However, Chile is implementing new strategies in this topic and created the “Road Coexistence Law” (which aims to put all modes of commuting on the roads on an equal footing), which came into effect from the end of 2018 and projects its effects towards 2030 [50]; therefore, it is expected that in some years, this area will improve. Finally, age was associated with active commuting behaviors. Older men and women students were more prone to use active commuting than young students. In Chile, it was found that active commuting decreases from high school to the university, giving space to more passive modes of commuting [36], which may be reflected the low adherence to active commuting in young students in the current results who have recently left high school.

In respect to the measurement of commuting behaviors, a study of 49 countries on five continents, including Chile, indicated that it is not possible to assess to what extent active commuting is contributing to the overall PA if commuting behaviors are not measured (e.g., time, distance, speed, and energy expenditure) [23]. These values are key to quantify the mode of commuting. Likewise, a systematic review in children and adolescents suggested that future studies are needed using a standardized self-report or objective measure that could accurately assess the characteristics of the mode of commuting to determine consistently the effect of active commuting [24]. In the current study, variables were calculated to provide further insights into the commuting behaviors. Secondly, to provide a deeper comprehension, the associations of the more standard categorical mode of commuting variable (active, public, and private) with each of the variables were conducted. The results showed that if the commuting time and distance increased, active commuting decreased in both men and women. According to this, it has been found that long commuting time is associated with public and private mode of commuting [51]. Moreover, longer distances have been presented in previous studies as the main barrier to being active commuters in children and adolescents [52] and university students [27,53]. Furthermore, the evidence presented (based on commuting speed and time) showed that energy expenditure per min in active mode of commuting involved the highest levels, followed by public and private mode of commuting, as also found in [40]. The total energy expenditure was higher in public commuting; this behavior is mostly in a seated state. Therefore, the energy expenditure per min is more representative and important regarding the trip. Consequently, choosing one active mode of commuting to and from university can make a large difference in the annual energy expenditure, as concluded in [26]. The current data concur with a study carried out in an English adult population that showed that active and public mode of commuting to work were important contributors to PA levels [54]. In addition, a study in university students from The United States showed that an increase of the energy expenditure per day, week, and year may result in gradual and sustained long-term improvements in cardiometabolic health [55]. In the same way, the Compendium of Physical Activities has received widespread acceptance as a resource to estimate and classify the energy cost of human PA [41]. Therefore, calculating the energy expenditure and quantifying new variables of commuting behaviors may provide a valuable contribution to more deeply understand the multiple benefits of active commuting modes to university or other frequent destinations, such as school or work. In addition, self-reported measures are a feasible and inexpensive way for easy dissemination and promotion among educational establishments and public institutions. However, more studies are required to examine these measurements in detail and establish a validation with objective measures.

Finally, in relation to commuting behaviors associated with sociodemographic characteristics, several sociodemographic characteristics have been shown to be strongly associated with active commuting behaviors in university students in the current study. The first strong predictor was the type of university and the geographical area that the students attended. Men and women who belonged to public university from Valparaíso were more likely to use active and public mode of commuting than students who belonged to private university from Santiago. Similar, a Spanish adolescents attending public schools had three and half times higher odds for use of active commuting compared with the adolescents attending private schools [56]. Actually, the type of university is a relevant socioeconomic indicator regarding the high difference between registration fees in Chilean public universities (e.g., cost of US $180) and private universities (e.g., cost of US $840). Secondly, there was a clear influence of the type of residence where the university students live in the commuting behaviors adopted. In the same way, it is important to highlight that choosing active commuting to university according to the geographical area may depend on specific characteristics of cities (such as environmental context, safety, orography, surface area, and population density). Therefore, each city must be considered independently, even if they belong to the same country. Additionally, men and women students who lived in university residences were more likely to use active commuting than those who live in a family residence. According to this, different studies in the United States indicated that type of residence was a strong indicator of the choice of mode of commuting to university, where the students who lived in the university residence tended to use active commuting [28,57]. The distance from the residence to university is a first predictor of active commuting modes to university [33,58], and it may be a factor explaining this difference regarding the type of residence. Actually, in the current study, the average distance to university of the students living in university residences was shorter (4.4 ± 6.6 km) than that for those living in family residences (13.5 ± 13.9 km). Finally, age and the socioeconomic level of the university students was associated with active commuting behaviors. Older men and women students were more prone to use active commuting than young students. It has been found that active commuting decreases from high school to the university, giving space to more passive modes of commuting [35,59], which may be reflected in the low adherence to active commuting in young students in the current results. In addition, men who belonged to low socioeconomic level and women who belong to medium and low socioeconomic levels preferred more active commuting modes than students who belonged to high socioeconomic level. The research indicated that investigating about this topic is necessary in the university population [27]. In the context of Saudi Arabia, university students who using non-motorized modes had a lower income range [17]. In Brazil, active commuting was associated with low income in work population [29], which is found as well in the United States among adolescents [60]. Therefore, it is necessary to further research into commuting behaviors, and their relationship with sociodemographic characteristics should be considered for future promotions of active commuting behaviors and generating a greater instance of PA in university students through choice of the mode of commuting.

### Limitations and Strengths

This study presents some limitations, for instance, the use of self-reported questionnaire and the lack of anthropometric parameters, which restricts the potential accuracy of the observed relationships in the present paper. The number of participants included was not representative of all the Chilean university students’ population, and it limited generalizability of findings to other parts of Chile. In addition, for the commuting distance self-reported, the shortest network distance on foot in kilometers (km) was used, but that distance may not be the one the student uses. On the other hand, one remarkable strength is that it provides new findings in an understudied population and presents new evidence to commuting patterns to and from university in Chilean students. This information could contribute to devising strategies for improvements in relation to PA in university students through active commuting.

## 5. Conclusions

This study presents new evidence of commuting patterns to and from university in Chilean students. In this study, the sociodemographic characteristics with the greatest influence over choice of active commuting to university were type of university, geographical area, type of residence, age, and socioeconomical level. The new variables of commuting behaviors calculated (speed and energy expenditure), based in self-reported variables (time and distance), provided valuable measurement information. However, further studies that include other new sociodemographic factors and objective measurements of commuting behaviors are required to examine more deeply this source of PA among university students.

## Figures and Tables

**Figure 1 ijerph-18-12557-f001:**
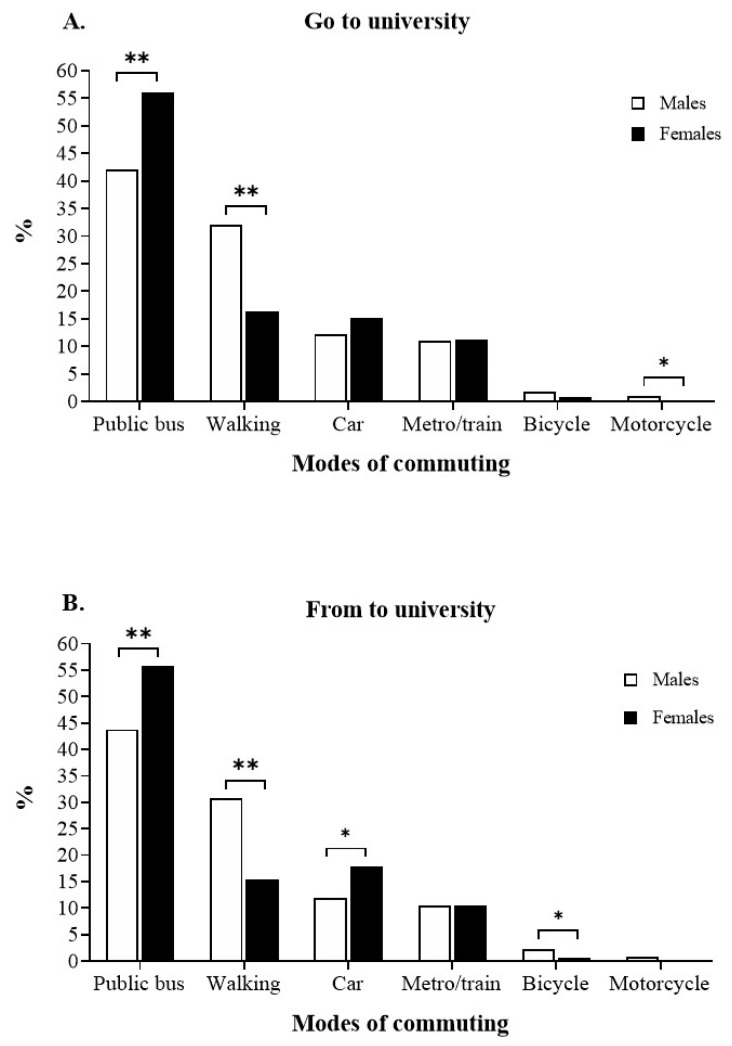
Commuting patterns both (**A**) to university and (**B**) from university in university students by gender. (*) Significant differences with *p* < 0.05 and (**) significant differences with *p* < 0.001.

**Table 1 ijerph-18-12557-t001:** Sociodemographic characteristics and commuting behaviors of participants by gender.

	All (*n* = 1257) X ± SD/*n* (%)	Men (*n* = 598) X ± SD/*n* (%)	Women (*n* = 659) X ± SD/*n* (%)	*p*-Value
Sociodemographic Characteristics				
Age (years)	22.4 ± 5.6	25.0 ± 7.2	20.2 ± 1.9	<0.001
Younger (18–25)	1021 (81.2)	384 (64.2)	637 (96.7)	<0.001
Older (>26)	236 (18.8)	214 (35.8)	22 (3.3)	
Type of residence				
Family residence	894 (71.1)	367 (61.4)	527 (20.0)	<0.001
University residence	363 (28.9)	231 (38.6)	132 (80.0)	
Locality area				
Rural	40 (3.2)	15 (2.5)	25 (3.8)	0.195
Urban	1217 (96.8)	583 (97.5)	634 (96.2)	
Socioeconomic levels				
High	213 (16.9)	92 (15.4)	121 (18.4)	0.356
Medium	661 (52.6)	318 (53.2)	343 (52.0)	
Low	383 (30.5)	188 (31.4)	195 (29.6)	
Type of university and geographical area				
Private (Santiago)	534 (42.5)	171 (28.6)	363 (55.1)	<0.001
Public (Valparaíso)	723 (57.5)	427 (71.4)	296 (44.9)	
Commuting Behaviors				
Mode of commuting				
Private	135 (10.7)	60 (10.0)	75 (11.4)	<0.001
Public	791 (62.9)	332 (55.5)	459 (69.7)	
Active	331 (26.3)	206 (34.4)	125 (19.0)	
Commuting time (min)				
Private	17.6 ± 22.0	18.5 ± 22.2	17.0 ± 21.8	<0.001
Public	24.2 ± 27.6	21.4 ± 27.2	26.7 ± 27.2	<0.001
Active	24.4 ± 17.9	25.0 ± 17.6	23.7 ± 18.2	<0.001
Commuting distance (km)				
Private	12.5 ± 16.7	11.0 ± 9.9	13.7 ± 20.6	<0.001
Public	14.4 ± 12.9	14.0 ± 13.2	14.7 ± 12.7	<0.001
Active	1.8 ± 3.3	1.8 ± 3.7	1.9 ± 3.7	<0.001
Commuting speed (km/h)				
Private	79.1 ± 125.5	57.7 ± 70.8	92.3 ± 148.3	<0.001
Public	28.4 ± 28.2	26.6 ± 23.7	29.7 ± 31.0	<0.001
Active	4.5 ± 8.5	3.2 ± 6.3	6.2 ± 10.4	<0.001
Energy expenditure per min (METs)				
Private	1.3 ± 0.0	1.3 ± 0.0	1.3 ± 0.0	0.985
Public	1.8 ± 0.4	1.8 ± 0.4	1.8 ± 0.5	0.122
Active	5.9 ± 2.4	5.8 ± 2.3	6.0 ± 2.6	0.096
Total Energy expenditure (METs)				
Private	70.9 ± 69.2	74.4 ± 77.7	67.4 ± 61.8	<0.001
Public	154.9 ± 66.4	149.9 ± 63.8	158.5 ± 67.4	<0.001
Active	120.5 ± 93.2	113.8 ± 99.6	131.7 ± 91.5	<0.001

Abbreviations: X ± SD, Mean ± Standard Deviation; min, minutes; km, kilometres; km/h, kilometres/hours; METs, Metabolic Equivalents.

**Table 2 ijerph-18-12557-t002:** Associations between mode of commuting with sociodemographic characteristics and commuting behaviors measures in university students by gender.

	Mode of Commuting to University **			
	Men		Women	
	Active	Public	Active	Public
	OR (95% CI)	OR (95% CI)	OR (95% CI)	OR (95% CI)
Sociodemographic				
Age (years)				
Younger (18–25)	Ref.	Ref.	Ref.	Ref.
Older (>26)	3.63 * (1.63, 8.10)	2.24 * (1.13, 4.44)	8.84 * (3.94, 13.78)	0.84 (0.21, 1.74)
Type of residence				
Family residence	Ref.	Ref.	Ref.	Ref.
University residence	12.43 * (4.39, 35.19)	2.33 (0.86, 6.25)	3.95 * (1.31, 11.85)	1.34 (0.50, 3.57)
Locality area				
Rural	Ref.	Ref.	Ref.	Ref.
Urban	1.57 (0.06, 36.43)	1.66 (0.20, 13.70)	1.47 (0.04, 44.94)	1.19 (0.32, 4.40)
Socioeconomic levels				
High	Ref.	Ref.	Ref.	Ref.
Medium	0.81 (0.31, 2.14)	1.30 (0.62, 2.73)	2.44 * (1.02, 5.81)	2.75 * (1.56, 4.83)
Low	3.10 (0.95, 10.09)	3.82 * (1.43, 10.18)	4.93 * (1.63, 14.90)	4.70 * (2.19, 10.06)
Type of university and geographical area				
Private (Santiago)	Ref.	Ref.	Ref.	Ref.
Public (Valparaíso)	26.75 * (10.63, 67.34)	13.68 * (6.34, 29.51)	8.02 * (3.00, 21.48)	3.75 * (1.74, 8.11)
Commuting Behaviors				
Commuting time	0.93 * (0.87, 0.99)	1.00 (0.94, 1.06)	0.92 * (0.88, 0.96)	0.99 (0.96, 1.02)
Commuting distance	0.54 * (0.31, 0.92)	1.02 (0.99, 1.05)	0.49 * (0.42, 0.57)	1.00 (0.98, 1.02)
Commuting speed	2.11 (0.16, 4.38)	0.94 (0.76, 1.15)	0.96 (0.91, 1.02)	0.97 (0.92, 1.02)
Energy expenditure per min	16.68 * (7.73, 19.69)	9.11 * (6.85, 15.07)	19.04 * (8.18, 25.68)	8.76 * (4.11, 13.20)
Energy expenditure	1.01 * (1.00, 1.02)	1.02 * (1.01, 1.03)	1.02 * (1.01, 1.03)	1.03 * (1.02, 1.04)

Analysis were adjusted for distance, age, and socioeconomic levels (except in the analysis when that variable was the predictor variable). Abbreviations: OR, Odd Ratio; 95% CI, 95% Confidence Intervals. (*) Significant association with *p* < 0.05; (**) Private commuting was stablished as reference.

**Table 3 ijerph-18-12557-t003:** Associations between commuting behaviors with sociodemographic characteristics in university students by gender.

	Commuting Behaviors to University			
	Men		Women	
	Speed	EE per min	Speed	EE per min
	*Beta* (95% CI)	*Beta* (95% CI)	*Beta* (95% CI)	*Beta* (95% CI)
Sociodemographic				
Age (years)				
Younger (18–25)	Ref.	Ref.	Ref.	Ref.
Older (>26)	−2.24 (−7.60, 3.11)	0.33 (−0.07, 0.73)	3.99 (−11.95, 19.94)	0.47 (−0.40, 1.35)
Type of residence				
Family residence	Ref.	Ref.	Ref.	Ref.
University residence	−19.64 * (−24.81, −14.47)	2.55 * (2.19, 2.90)	−19.12 * (−25.72, −12.52)	2.05 * (1.68, 2.43)
Locality area				
Rural	Ref.	Ref.	Ref.	Ref.
Urban	15.21 (−4.25, 34.68)	−1.22 (−2.50, 0.05)	24.97 * (6.06, 43.87)	−0.93 (−1.76, −0.11)
Socioeconomic levels				
High	Ref.	Ref.	Ref.	Ref.
Medium	1.46 (−2.56, 5.49)	0.07 (−0.22, 0.37)	4.27 (−0.99, 9.56)	−0.01 (−0.27, 0.19)
Low	−1.67 (−5.46, 2.16)	0.01 (−0.19, 0.22)	−4.18 (−8.77, 0.41)	0.76 (−0.19, 0.34)
Type of university and geographical area				
Private (Santiago)	Ref.	Ref.	Ref.	Ref.
Public (Valparaíso)	−10.03 * (−17.13, −2.93)	1.34 * (0.92, 1.77)	−5.65 (−13.09, 1.88)	0.97 * (0.62, 1.33)

Analysis were adjusted for age and socioeconomic levels (except in the analysis when that variable was the predictor variable). Abbreviations: Beta, Unstandardized Beta coefficient; 95% CI, 95% Confidence Intervals; EE, Energy expenditure; * Significant differences with *p* < 0.05.

## Data Availability

All data are available on paper in the laboratory of the IRYS research group, at the School of Physical Education in Viña del Mar, Valparaíso, Chile.

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
