# Peer review of "New Self-Report Measures of Commuting Behaviors to University and Their Association with Sociodemographic Characteristics"

_ijerph, 2021, doi:10.3390/ijerph182312557_

Round 1

Reviewer 1 Report

Dear Authors, the topic of the study is important from the point of view of practical application for health promotion. The research is done very carefully, the results thoroughly analyzed and well described. The use of energy expenditure proposed by you to assess commuting behaviors is not a new idea, while the assessment of speed is an interesting idea, but its implementation is very time-consuming. It would be worth checking in the future what new information it provides and whether this data contributes to the risk assessment of future health problems. The publication lacks data on basic parameters, such as BMI, which seems to be directly related to physical activity. The discussion lacks an attempt to explain why older students have better commuting behaviors compared to younger students, it seems that the opposite should be the case.

Author Response

We would like to thank the reviewers for their thoughtful and constructive comments and feedback. We have considered all the suggestions and have incorporated them into the revised manuscript.

Please, find a new revision of our manuscript “New self-report measures of commuting behaviors to university and their association with sociodemographic characteristics", in addition to the responses to your suggestions that are attached. We believe our manuscript is stronger because of these revisions.

Cordially,

The authors.

Reviewer 2 Report

The article deals with a descriptive study of the to and from active commuting the university of students and examines the sociodemographic characteristics associated with displacement and gender. While this topic seems to be of importance, there are a few revisions that I recommend before accepting this document for publication. The following are section-specific questions, comments, and concerns.

Correct the word behaviours by behaviors

Introduction:

The introduction mentions the association between physical inactivity and its negative repercussions on people's quality of life, as well as its possible influence on non-communicable diseases. It also refers to the promotion of physical activity through active travel, such as walking or cycling, as a simple and easy way to incorporate them into daily routines. Some studies carried out in others are mentioned. Although they mention studies carried out in other countries, the characteristics of the countries are not addressed (for example a friendly city with active commuting). It is recommended to mention some data of the country and if studies have been carried out before.

Method:

Better describe the instrument (total number of questions). Was a questionnaire validation process carried out?

I believe that the variable of the cities where the questionnaire was applied has not been considered, since Chile has a diverse orography and population among its cities and can determine the use of the commuting active

Results

The results are presented correctly. But I recommend doing an analysis of the results according to the city.

Discussion:

Although the discussion is supported by other studies from other countries, these have different realities and do not delve into the causes that may be behind the results, such as the surface of the city, the existence of bicycle lanes .....

Author Response

(The authors gave the same response as above.)

Round 2

Reviewer 2 Report

Thank you for attending to the suggestions indicated. However, on line 22, you mention that the students come from three universities in different cities.  As I have indicated previously, it is important to know the characteristics of the cities, since it can be an important variable that can influence the results. The authors have followed most of the indications, but they have not indicated the characteristics of the cities where the study was done out, being an important variable of information to understand the results, since Chile has a very diverse orography, surface area, and population density.

Author Response

Please, find a new revision of our manuscript “New self-report measures of commuting behaviors to university and their association with sociodemographic characteristics”. We would like to thank for the thoughtful and constructive comment and feedback. We have considered the new suggestion and have incorporated them into the revised manuscript. All changes to the original manuscript that you find are in relation to this new suggestion.

Cordially, the authors.
